# Genetic Diversity of *Plasmodium falciparum* in Korhogo Health District, Northern Côte d’Ivoire

**DOI:** 10.3390/tropicalmed10090255

**Published:** 2025-09-07

**Authors:** Edjronké M. A. Benié, Carla Beuret, Annina Schnoz, Sara L. Cantoreggi, Xavier C. Ding, Kigbafori D. Silué, Christian Nsanzabana

**Affiliations:** 1Laboratoire de Biologie et Cytologie Animale, Unité de Formation et de Recherche Sciences de la Nature, Université Nangui Abrogoua, Abidjan 02 BP 801, Côte d’Ivoire; 2Centre Suisse de Recherches Scientifiques en Côte d’Ivoire, Abidjan 01 BP 1303, Côte d’Ivoire; kigbafori.silue@csrs.ci; 3Swiss Tropical and Public Health Institute, Kreuzstrasse 2, 4123 Allschwil, Switzerland; carla.beuret@swisstph.ch (C.B.); annina.schnoz@swisstph.ch (A.S.); sara.cantoreggi@swisstph.ch (S.L.C.); christian.nsanzabana@swisstph.ch (C.N.); 4University of Basel, Petersplatz 1, 4001 Basel, Switzerland; 5FIND, Campus Biotech, 9 Ch. des Mines, 1202 Geneva, Switzerland; xavier.ding@gmail.com; 6Laboratoire de Biologie et Santé, Unité de Formation et de Recherche Biosciences, Université Félix Houphouët-Boigny, Abidjan 01 BP V34, Côte d’Ivoire

**Keywords:** *Plasmodium falciparum*, *msp1*, *msp2*, *glurp*, Korhogo, Côte d’Ivoire

## Abstract

Understanding *Plasmodium falciparum* population genetic diversity is crucial to assess the impact of malaria control interventions. This study investigated *P. falciparum* genetic diversity using merozoite surface protein 1 (*msp1*), *msp2* and glutamate-rich protein (*glurp*) in Korhogo district, Northern Côte d’Ivoire. DNA was extracted from dried blood spots (DBSs) collected in the health district of Korhogo between 2019 and 2020. The *msp1*, *msp2*, and *glurp* genes were amplified by polymerase chain reaction (PCR), and amplicon sizes were determined by capillary electrophoresis. Out of 179 samples randomly selected and genotyped, 82% were successfully amplified for *msp1*, 85% for *msp2*, and 75% for *glurp*. For *msp1*, the K1 allelic family had 20 genotypes, MAD20 had 23, and RO33 had only one genotype. For *msp2*, there were 59 and 33 genotypes for 3D7 and FC27, respectively, and for *glurp*, 45 genotypes were detected. The parasite population was highly diverse with an expected heterozygosity (H_E_) of ≥0.9 for all 3 markers. Our study showed high genetic diversity of *msp1*, *msp2*, and *glurp* in *P. falciparum* isolates from Korhogo district, Northern Côte d’Ivoire. These data could provide baseline information on *P. falciparum* genetic diversity for further epidemiological studies, needed to assess interventions implemented in this area.

## 1. Introduction

Malaria remains a major public health issue in Côte d’Ivoire. Despite control efforts over the last decades, malaria incidence increased in the country from 155 in 2016 to 191/1000 people at risk in 2020 [1]. However, the malaria mortality rate decreased by 59% in the same period [1]. The disease poses a heavy burden on the health system at the national level, with a prevalence estimated at 33% in 2022 [2]. More than 94% of malaria infections in Côte d’Ivoire are due to *Plasmodium falciparum* [3]. This deadly species is also associated with antimalarial drug resistance [4].

Malaria control strategies in Côte d’Ivoire are based on (i) malaria prevention through long-lasting insecticidal nets (LLINs); (ii) intermittent preventive treatment during pregnancy (IPTp); and (iii) prompt diagnosis and effective treatment of clinical cases with Artemisinin-based Combination Therapies (ACTs) [2]. In 2021, the fourth national campaign, organized by the National Malaria Control Program (NMCP), enabled the distribution of 19 million LLINs, with 80% utilization as an achievement [1]. Routine monitoring of ACTs efficacy for first- and second-line treatment of *P. falciparum* malaria is being conducted on a routine basis [4]. In 2023, seasonal malaria chemoprevention (SMC) was introduced in the Northern part of the country, and the R21/Matrix-M malaria vaccine was introduced in 2024 for children aged 6 to 23 months [5]. Moreover, Perennial Malaria Chemoprevention (PMC) is being piloted in the regions not implementing SMC [6].

For targeted and tailored control interventions, a better understanding of *P. falciparum* population genetics is a prerequisite. Parasite genomic data can provide detailed information on parasite connectivity and transmission to guide intervention planning and assess the impact of interventions towards reducing malaria transmission [7]. Population genetic methods provide powerful means to monitor changes in the parasite population related to transmission and evaluate the potential risk of drug resistance selection on that population [8].

The merozoite surface proteins 1 (*msp1*), and 2 (*msp2*) and glutamate-rich protein (*glurp*) are commonly used for *P. falciparum* genotyping, to assess parasite population genetics, and potentially the impact of interventions on the parasite population genetics [9]. Moreover, they are used in anti-malarial drug efficacy trials to distinguish recrudescent parasites from new infections, with glurp being recently replaced by microsatellites [10]. *Msp*/*glurp* typing is particularly useful in determining the multiplicity of infection (MOI), which has been suggested as a proxy for local transmission intensity [11].

*Msp*/*glurp* markers have been previously used by few studies for *P. falciparum* genotyping in Côte d’Ivoire to distinguish recrudescence from new infection [4]. The available evidence about the parasites population genetics using *msp*/*glurp* markers is mainly from the Western, Southern and Eastern parts of the country, and all studies have reported high genetic diversity [12,13,14]. There is no data on the *P. falciparum* parasites population in the Northern part of the country. However, this area is characterized by seasonal malaria transmission, and has recently introduced SMC [15]. This study investigated *P. falciparum* genetic diversity using *msp1*, *msp2*, and *glurp* in Korhogo district, Northern Côte d’Ivoire. The data generated could provide baseline information on *P. falciparum* genetic diversity for further epidemiological studies, needed to assess interventions implemented in this area.

## 2. Materials and Methods

### 2.1. Study Site

Korhogo is located in the Northern part of Côte d’Ivoire. This area is in the Southern part of the Sahelian band. The climate is characterized by two seasons: a rainy season that occurs from May to October and a dry season from November to April. The average annual temperature is 27.0 °C, while annual rainfall varies from 1200 to 1400 mm. The population was estimated in 2021 at 440,926 inhabitants. Korhogo is one of the sentinel sites for therapeutic efficacy studies (TES) in Côte d’Ivoire [2]. This site records low-intensity malaria transmission throughout the year. Between 2004 and 2013, the incidence of malaria was highly seasonal [16]. During the same period, 198,679 households were counted and 163,302 LLINs were distributed [17]. Another study revealed that LLINs were used by 61.4% of households [18]. Between 2019 and 2020, malaria prevalence was 14% [19], and the area was selected in 2021 by NMCP for piloting SMC implementation [15].

### 2.2. Sample Collection

Samples were collected from August 2019 to February 2020, through one cross-sectional survey from 3 urban health centers and households surrounding them (Centre de santé urbain (CSU) Petit Paris, CSU Tchelelezo and CSU Kokoton), and one rural health center (Centre de santé medico-social (CMS) Torgokaha); the latter being 8 km from the city center (Figure 1). Briefly, study participants were recruited through passive case detection (PCD) at point-of-care and active case detection (ACD) during household visits. The ACD approach involved health centers, and was conducted in households around these health centers. Febrile patients and asymptomatic participants were screened according to the following inclusion criteria. Febrile patients screened were (i) age ≥ 10; (ii) fever (axillary temperature ≥ 37.5 °C) or self-reporting history of fever during the last 48 h; (iii) absence of severe pathology or infection of central nervous system; (iv) no self-reported use of antimalarial treatments over the four previous weeks; (v) no pregnancy (confirmed or suspected). Asymptomatic participants were (i) age ≥ 10; (ii) absence of fever (axillary temperature < 37.5 °C) or self-reporting history of fever during the last four weeks; (iii) no pregnancy (confirmed or suspected). For both categories (febrile or asymptomatic), we obtained written informed consent from adult participants and assent from participants aged 10 to 17 years old. Prior to the assent, written informed consent was also obtained from legal representative of minor participants.

Capillary blood samples were collected for malaria diagnosis using HRP2-based rapid diagnostic test (RDT) and microscopy. Finger-prick blood was collected on Whatman 903^TM^ DNA, and Whatman FTA^TM^ Mini Card, GE Healthcare Ltd., Cardiff, UK, dried and stored in individual plastic zip bags with desiccant, at room temperature. DBS were transferred to the Swiss Tropical and Public Health institute (Swiss TPH), Allschwil, Switzerland, for molecular analysis.

### 2.3. DNA Extraction

Genomic DNA was extracted from DBS on Whatman 903^TM^ DNA, and Whatman FTA^TM^ Mini Card, using QIAamp DNA Mini Kit, QIAGEN GmbH, Hilden, Germany, according to the manufacturer’s instructions [20]. Genomic DNA was isolated from 3 punches of 3 mm diameter. Extracted DNA was eluted in 100 µL of elution (AE) buffer and stored at −20 °C until analysis.

### 2.4. Plasmodium falciparum Confirmation and Quantification

Quantitative PCR (qPCR) were performed to confirm and quantify *P. falciparum* infections using *P. falciparum* var acidic terminal sequence (varATS) as described elsewhere [21]. Each assay was run in duplicate including Non-Template Control (NTC) and 3D7 strains tenfold diluted ranged from 47,000 to 0.47 parasites/µL for the standard curve.

### 2.5. Msp1, Msp2, and Glurp Molecular Analysis

The *msp1*, *msp2,* and *glurp* PCR amplifications were carried out as previously described [22]. PCR amplification was confirmed by QIAxcel Advanced System^®^ from Qiagen.

All positive samples for *Pfmsp1*, *Pfmsp2*, and *Pfglurp* were prepared for capillary electrophoresis except those positive for RO33 allele of *Pfmsp1*. The DNA concentration of the nested PCR products was determined using the Synergy H1 microplate reader, BioTek Instruments, Inc., Winooski, VT, USA, and Promega QuantiFluor^®^ dsDNA System kit, Promega Corporation, Madison, WI, USA. The nested PCR products were then diluted with Milli-Q water using QIAGEN QIAgility version 4.17.1 to a target concentration depending on the marker. The final concentration range established for each allelic family was 0.5 ng/μL for 3D7 and FC27, 0.8 ng/μL for K1, 1.5 μg/uL for MAD20, and 3 ng/μL for *glurp*. Two microliters of diluted nested PCR product were mixed with 12 μL of Hi-Di™ Formamide and 0.2 μL of GeneScan™ 500 LIZ^®^ size standard (for all *msp1 and msp2* allelic families except RO33 of *Pfmsp1*) or 1200 LIZ^®^ size standard (for *glurp*). Samples were transferred into MicroAmp^®^ Optical 96-Well reaction plates and analyzed with an automated sequencer (Applied Biosystems AB31370xl at Microsynth AG, Balgach, Switzerland).

Sequencing data were analyzed by GeneMapper^®^ software version 6 (Applied Biosystems LTD, Framingham, MA, USA). Allele calling and sizing was performed as previously described [22]. For each *msp1*, *msp2* allelic families, and *glurp* gene, fragment sizes were assigned to specific genotypes using a bin size of 3 bp [23]. Genetic diversity pie charts were completed using R Studio 2022.02.3.

RO33 was not considered as a polymorphic marker, therefore a standard nested PCR was performed, and only the presence or absence of an amplicon was recorded.

### 2.6. Allele Frequency and Multiplicity of Infection

The frequency of each allelic family for *msp1* and *msp2* genes was determined as the proportion of successfully genotyped samples for a given allelic family out of overall positive samples for *msp1* or *msp2* genes corresponding [24]. MOI was defined as the number of distinct parasite genotypes co-existing within a given infection based on the genotyping of the *msp1, msp2*, and *glurp* genes. Briefly, MOI of a given sample is based on the MOI of *msp1*, *msp2*, and *glurp* genes. MOI for *msp1* resulted from the addition of K1, MAD20, and RO33’s genotypes numbers; while *msp2* MOI was obtained by summing the number of genotypes of 3D7 and FC27; and *glurp*’s MOI corresponded to the number of genotypes for this gene. So, the overall MOI was the MOI of the gene with the highest value [25]. Samples with a single parasite genotype were classified as mono-infection while samples with more than one parasite genotype were classified as polyclonal infection [24].

### 2.7. Heterozygosity

Population-level genetic diversity was assessed by expected heterozygosity (H_E_). It was calculated as previously described [26]. Briefly, H_E_ was calculated for each locus using the equation H_E_ = [n/(n − 1)] [1 − ∑pi^2^], where n is the number of samples analyzed, pi is the allele frequency of the allele in the population. H_E_ ranges from 0 to 1 (0 indicating no diversity and 1 indicating all alleles are different). Mean H_E_ was calculated by taking the average of H_E_ across all loci.

### 2.8. Statistical Analysis

Statistical analysis were performed using Stata (version 16.1) software. Prevalence of polyclonal infection was determined with 95% confidence interval (95% CI). Pearson chi-square or Fisher’s exact tests were used to establish the associations between age, sex, status, site, approach (PCD or ACD), and polyclonal infections. The MOI between groups was compared using a Mann–Whitney test. The Spearman’s rank correlation coefficient was calculated to assess relationship between MOI and parasite density or age. *p* values less than 0.05 were considered statistically significant.

### 2.9. Ethics

This study was approved in 2019 by the Comité National d’Ethique des Sciences de la Vie et de la Santé of Côte d’Ivoire (CNESVS) (N/Réf: 027-19/MSHP/CNESVS-kp).

## 3. Results

### 3.1. Study Population

A total of 179 DBS samples positive for RDT or microscopy were selected for genetic diversity analysis. The characteristics of the study population are presented in Table 1. The mean age was 23.2 years (standard deviation (SD): 12.5) with minimum and maximum ages of 10 and 67 years, respectively. The proportion of adults (>15 years) was higher with 68% (122/179) compared to children. In total, 52% of participants were female (93/179), and 79% (141/179) of participants were from urban settings. Most of the participants with 93% (166/179) were febrile, while asymptomatic carriers represented only 7% (13/179). Overall, 60% (107/179) of the selected patients, were recruited through PCD, and the remaining 40% recruited through ACD. All samples were confirmed to be *P. falciparum*, and the parasite density mean was 2697.9 parasites/μL (95% CI 1454.5–3941.4).

### 3.2. Msp1, Msp2 and Glurp Distribution

*Msp1*, *msp2*, and *glurp* diversity profiles are shown in Table 2. Out of 179 samples analyzed, 82% (146/179) were successfully amplified for *msp1*, 85% (152/179) for *msp2*, and 75% (135/179) for *glurp*. For *msp1*, the K1 allelic family was predominant and was found in 72% (105/146) of samples, followed by the MAD20 allelic family in 40% (59/146) of samples, and the RO33 allelic family in 36% (52/146) of samples. For *msp2*, the prevalence of 3D7 and FC27 allelic families were 84% (127/152) and 64% (98/152), respectively.

### 3.3. Genetic Diversity and Allelic Frequency

The genetic diversity was high for all markers used, with the H_E_ being ≥0.9 for all markers (Table 2). The number of genotypes for each marker are shown in Table 2. There were 20 genotypes for K1, and 23 for MAD20 for *msp1* allelic families. The *msp2* allelic families harbored 59 and 33 genotypes for 3D7 and FC27, respectively. A total of 45 different *glurp* genotypes were detected.

The fragment sizes and frequencies of each marker are shown in Appendix A. For *msp1* allelic families, the fragment sizes ranged from 128 to 292 bp for K1 and 110 to 246 bp for MAD20. Most of these genotypes occurred at a frequency below 10% and only a few above. Three genotypes of K1 allelic family (230, 239 and 212 bp) and two genotypes of MAD20 allelic family (219 and 201 bp) had a frequency above 10% (Appendix A). For *msp2* allelic families, the fragment sizes ranged from 169 to 575 bp for 3D7 and 171 to 619 bp for FC27. All 3D7 genotypes and a high proportion of FC27 genotypes occurred at a frequency below 10%. However, 3 FC27 genotypes (335 bp, 371 bp, 418 bp) occurred at a frequency above 10% (Appendix A). In *glurp*, the fragment sizes ranged between 491 and 1159 bp. All genotypes occurred at frequencies below 10% (Appendix A).

### 3.4. Multiplicity of Infection

Out of 179 samples, 88% (157/179) yielded a positive result for either *msp1*, *msp2*, or *glurp* genes. For these all-positive samples, the mean MOI was 1.9 (CI95%, 1.8–2.1) with two-thirds (67%) harboring polyclonal infections. The MOI was 1.5 for *msp1*, 1.8 for *msp2*, and 1.2 for *glurp.* The prevalence of polyclonal infections was 71% (103/146) for *msp1*, 68% (104/152) for *msp2*, and 72% (97/135) for *glurp* (Table 3).

Polyclonal infections and MOI across age, sex, health status, site, and recruitment approach are shown in Table 4. The proportion of isolates with polyclonal infection was high (>65%), and did not differ significantly between the different groups. However, the proportion of polyclonal infections was significantly higher in the urban area (71.4%; 90/126), compared to the rural area (48.4%, *p* = 0.015). Also, a higher value of parasite density in the urban area compared to the rural area was found (3721 vs. 452.5 parasites/μL, z = 2.743, *p* = 0.006) (Appendix A). The MOI varied from 1.2 to 2 across age, sex, health status, and recruitment approach, but no significant difference was found between different groups. The MOI was correlated with parasite density (r = 0.3861, *p* < 0.001), not with age (r = −0.0695, *p* = 0.3868) (Appendix A).

## 4. Discussion

The *msp1*, *msp2*, and *glurp* genes are used as markers to determine *P. falciparum* genetic diversity which are crucial to understand malaria transmission dynamics [9]. So, far, there was no data on *P. falciparum* genetic diversity in the Northern part of Côte d’Ivoire, and this is the first study to investigate *P. falciparum* genetic diversity using *msp1*, *msp2*, and *glurp* in *P. falciparum* isolates from Korhogo district.

In this study, K1 was the most abundant *msp1* allelic family, whereas 3D7 was the most frequent *msp2* allelic family, and this is in line with other studies that have assessed the genetic diversity of *P. falciparum* in the country [12,27]. Moreover, the same pattern is frequently reported throughout Sub-Saharan Africa, whereas in Asia, MAD20 was most frequent for *msp1* and for *msp2*, a similar prevalence of 3D7 and FC27 allelic families is often reported [28,29,30]. Similarly, *glurp* genetic diversity found in this study was high and similar to other studies in Sub-Saharan countries [28].

Using capillary electrophoresis, a high number of alleles for each gene was observed. A total of 20 genotypes were recorded for K1, 23 for MAD20, 59 for 3D7, 33 for FC27, and 45 for *glurp*. Compared to the previous studies based on the agarose gel electrophoresis, fewer genotypes were found in the country [12,27]. The high number of clones in the current study was due to the capillary electrophoresis method used to analyze amplified fragments. Capillary electrophoresis provides accurate and reproducible estimates of DNA fragment lengths with a resolution power down to a few base pairs difference [31].

In this study, RO33 was not included in the capillary electrophoresis analysis. RO33 allele is considered to be monomorphic with amplified fragment size around 155 bp [32].

Based on the high number of alleles, it is no surprise that this study recorded high values of H_E_. Indeed, *P. falciparum* isolates from Korhogo, are characterized by a high genetic diversity. This high genetic diversity is common in Sub-Saharan Africa, while an intermediate level is observed in Southeast Asia, and Latin America is characterized by low genetic diversity [33,34,35].

As generally observed in high transmission settings, the proportion of polyclonal infections was higher compared to monoclonal infections [8]. In this study, two-thirds of the isolates harbored more than one clone. This outcome is in line with other studies conducted in neighboring regions, namely South of Mali and South-West of Burkina Faso, which reported high proportions of polyclonal infection [36,37]. The MOI was found to be moderate in this region, with an average of 1.9 different clones per sample. This area’s MOI is lower than those reported across different regions in Côte d’Ivoire, ranging from 2.8 in the West and South to 4 in the East region [12,13,14]. Malaria transmission is high in Korhogo and surrounding areas, with an average entomological inoculation rate (EIR) of 94.9–242.49 infected bites per person per year [17].

In this study, the MOI was positively correlated with parasite density, MOI increasing with parasite density. It is in line with a previous study, in which higher MOI and parasitaemia were harbored by symptomatic individuals compared to asymptomatic [38]. However, in our study, similar MOI was found in asymptomatic carriers and malaria patients, but the proportion of asymptomatic carriers was too low to preclude any definitive conclusion.

In this study, polyclonal infection and MOI were higher in the urban area compared to the rural setting. Furthermore, the average parasite density was higher in the urban area compared to the rural setting. These findings suggest that malaria transmission is higher in Korhogo compared to surrounding rural settings. According to a previous study conducted in Korhogo, rural inhabitants are slightly less exposed to *Plasmodium*-infected *Anopheles* bites compared to the ones in the urban area [17], which is in line with our findings. It is possible that Korhogo city expansion and population migration from the area including a high malaria transmission setting could explain our findings. Indeed, in the last two decades, a greater increase in malaria transmission intensity in urban settings across sub-Saharan compared to rural areas was observed, and calls for heightened vigilance in malaria surveillance in urban settings [39].

The data generated in this study, such as MOI and H_E_ could serve as baseline for evaluating the impact of future malaria interventions in the area. For instance, the decline in MOI and H_E_ over time was associated with successful malaria intervention strategies in Ethiopia [40].

A weakness of this study was the variation in sample sizes biased towards symptomatic patients. Future studies should try to include a higher proportion of asymptomatic carriers that have been shown to carry a large reservoir of parasites [41].

## 5. Conclusions

*P. falciparum* isolates from Korhogo, in the Northern part of Côte d’Ivoire, displayed high genetic diversity and allelic frequency. Two-third of the samples harbored more than one *P. falciparum* clone. The higher proportion of polyclonal infections and higher parasite density in the urban areas compared to the rural areas warrants further investigation. Future studies should investigate the use of amplicon deep sequencing that could potentially give a more granular picture of the parasite genetic diversity in the region, and explore the prevalence of molecular markers of resistance.

## Figures and Tables

**Figure 1 tropicalmed-10-00255-f001:**
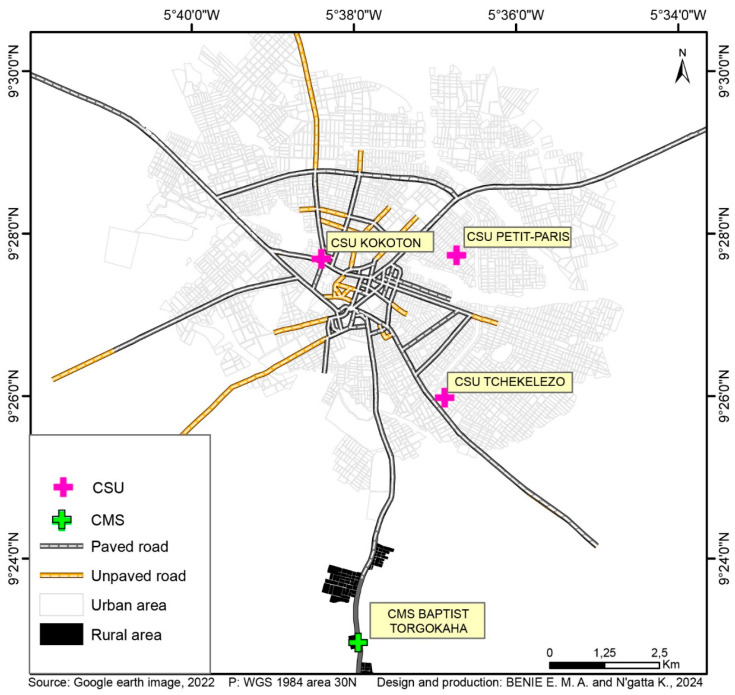
Study site.

**Table 1 tropicalmed-10-00255-t001:** Population characteristics.

	Sample, N = 179
Parasite density mean (parasites/μL blood)	2697.9 (95% CI 1454.5–3941.4)
Age mean ± SD (Range)	23.2 ± 12.5 (10–67)
Age groups	
≤15 years, n (%)	57 (32)
>15 years, n (%)	122 (68)
Sex	
Male, n (%)	86 (48)
Female, n (%)	93 (52)
Site	
Urban, n (%)	141 (79)
Rural, n (%)	38 (21)
Status	
Asymptomatic, n (%)	13 (7)
Febrile, n (%)	166 (93)
Approach	
PCD, n (%)	107 (60)
ACD, n (%)	72 (40)

N, number of samples examined; CI, confidence interval; SD, standard deviation; n, number; PCD, passive cases detection; ACD, active cases detection.

**Table 2 tropicalmed-10-00255-t002:** *Msp1*, *msp2*, and *glurp* diversity profiles.

	Positive, n/N (%)	Number of Genotypes	Expected Heterozygosity
*msp1*	146/179 (82)	-	-
K1	105/146 (72)	20	0.92
MAD20	59/146 (40)	23	0.91
RO33	52/146 (36)	1	-
*msp2*	152/179 (85)	-	-
3D7	127/152 (84)	59	0.98
FC27	98/152 (64)	33	0.90
*glurp*	135/179 (75)	45	0.96

n, number; N, number of samples examined; *msp1*, merozoite surface protein 1; *msp2*, merozoite surface protein 2; *glurp*, glutamate-rich protein.

**Table 3 tropicalmed-10-00255-t003:** Polyclonal infection and MOI of *msp1*, *msp2*, and *glurp*.

	Monoclonal Infections n/N (%)	Polyclonal Infections n/N (%)	MOI Geometric Mean (95% CI)
*msp1*	43/146 (29)	103/146 (71)	1.5 (1.4–1.7)
*msp2*	48/152 (32)	104/152 (68)	1.8 (1.6–1.9)
*glurp*	38/135 (28)	97/135 (72)	1.2 (1.1–1.3)
Total	52/157 (33)	105/157 (67)	1.9 (1.8–2.1)

n, number; N, number of samples examined; MOI, multiplicity of infection; CI, confidence interval; *msp1*, merozoite surface protein 1; *msp2*, merozoite surface protein 2; *glurp*, glutamate-rich protein.

**Table 4 tropicalmed-10-00255-t004:** Polyclonal infection and MOI across age, sex, status, site, and approach.

	Total, N	Polyclonal, n (%)	95% CI	*p* Value	MOI Geometric Mean (95% CI)	*p* Value
All	157	105 (66.9)	59.4–74.3		1.9 (1.8–2.1)	NA
Age						
≤15 years	50	33 (66)	52.4–79.6		1.9 (1.6–2.3)	
>15 years	107	72 (67.3)	58.2–76.3	0.873	1.9 (1.7–2.1)	0.987
Sex						
Male	78	53 (67.9)	57.4–78.5		1.9 (1.7–2.1)	
Female	79	52 (65.8)	55.1–76.5	0.777	2.0 (1.7–2.2)	0.636
Status						
Asymptomatic	7	5 (71.4)	26.3–100		2.0 (1.1–3.6)	
Febrile	150	100 (66.7)	59–74.3	0.576	1.9 (1.8–2.1)	0.8
Site						
Rural	31	15 (48.4)	29.7–67		1.7 (1.4–2.2)	
Urban	126	90 (71.4)	63.4–79.4	0.015	2.0 (1.8–2.2)	0.146
Approach						
PCD	98	66 (67.3)	57.9–76.8		2.0 (1.8–2.2)	
ACD	59	39 (66.1)	53.7–78.5	0.872	1.9 (1.6–2.2)	0.6621

N, number of samples examined; n, number; CI, confidence interval; MOI, multiplicity of infection; PCD, passive cases detection; ACD, active cases detection.

## Data Availability

The datasets supporting the conclusions of this article are included within the article. Raw data used for analysis of this study are available from the corresponding author on reasonable request.

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
