# Peer review of "Genetic Diversity of Plasmodium falciparum in Korhogo Health District, Northern Côte d’Ivoire"

_tropicalmed, 2025, doi:10.3390/tropicalmed10090255_

Round 1
Reviewer 1 Report
Comments and Suggestions for Authors
Lines 98 and 102: age ≥ 10. Why younger than 10 years were excluded?
Lines 108-109: Capillary blood samples were collected for malaria diagnosis using HRP2-based rapid diagnostic test (RDT) and microscopy. Also finger-prick or using venipuncture and then apply anticoagulants? Also consider the methodology. As you mentioned in lines 179-180: were some samples excluded due to co-infections with other plasmodium species? False positive?
Line 116: instructions (20). Genomic DNA was isolated from 3 punches of 3mm diameter. No need for reference here, just mention the company and country of the kits manufacturer.
Line 127: All positive samples for Pfmsp1, Pfmsp2, and Pfglurp were prepared for capillary electrophoresis except those positive for RO33 allele of Pfmsp1. Why? And then in line 135 the authors stated this: (for all msp1 and msp2 allelic families). And again justified the reason for not including the RO33 in lines 144-145. Remain consistent.
lines 156-157: Samples with a single parasite genotype for all three genes were classified as mono-infections while samples with more than one parasite genotype with any of the three genes were classified as polyclonal infection. Revise the sentence as it states that you combined the genotyping of the gene. For instance, a sample positive for msp1 MAD20 and msp2 FC27 and glurp can not be consider polyclonal. Here its only monoclonal. Your sentence state otherwise.
Table 2. as mentioned, 146 successfully detected the msp1. This means 33 samples failed. Are they parasite negative samples? Also, why only consider the alleles frequencies and not showing the allelic families such as MAD20/K1 or MAD20/RO33 combinations. The authors already mentioned in lines 156-157: Samples with a single parasite genotype for all three genes were classified as mono-infections while samples with more than one parasite genotype with any of the three genes were classified as polyclonal infection.
Author Response
We would like to thank you for taking the time to review this manuscript. Indeed, you have carefully studied our manuscript according to your constructive comments and suggestions to improve this current paper and further studies. Please find the detailed responses below and the corresponding revisions/corrections highlighted/in track changes in the re-submitted files.
Comments 1: Lines 98 and 102: age ≥ 10. Why younger than 10 years were excluded?
Response 1: Age <10 is not included in the recruitment, as we needed a population that could: (i) understand and give their own approval; (ii) ability to better tolerate the discomfort associated with capillary blood sampling and further venous blood sampling planned by the initial project.
Comments 2: Lines 108-109: Capillary blood samples were collected for malaria diagnosis using HRP2-based rapid diagnostic test (RDT) and microscopy. Also finger-prick or using venipuncture and then apply anticoagulants? Also consider the methodology. As you mentioned in lines 179-180: were some samples excluded due to co-infections with other Plasmodium species? False positive?
Response 2: The samples collected in this study are capillary blood obtained from a finger-prick. All samples included were confirmed to be P. falciparum using qPCR based on Var ATS, so none was excluded.
Comments 3: Line 116: instructions (20). Genomic DNA was isolated from 3 punches of 3mm diameter. No need for reference here, just mention the company and country of the kits manufacturer.
Response 3: The Filter paper used are manufactured by GE Healthcare Ltd, UK . It was added in the revised version (see revised manuscript line 116).
Comments 4: Line 127: All positive samples for Pfmsp1, Pfmsp2, and Pfglurp were prepared for capillary electrophoresis except those positive for RO33 allele of Pfmsp1. Why? And then in line 135 the authors stated this: (for all msp1 and msp2 allelic families). And again justified the reason for not including the RO33 in lines 144-145. Remain consistent.
Response 4: RO33 was not included in the capillary electrophoresis analysis as it is considered to be monomorphic. In order to be consistence, the sentence ‘for all msp1 and msp2 allelic families’ has been rewritten by adding ‘except RO33 of Pfmsp1’ (see revised manuscript line 138).
Comments 5: lines 156-157: Samples with a single parasite genotype for all three genes were classified as mono-infections while samples with more than one parasite genotype with any of the three genes were classified as polyclonal infection. Revise the sentence as it states that you combined the genotyping of the gene. For instance, a sample positive for msp1 MAD20 and msp2 FC27 and glurp can not be consider polyclonal. Here its only monoclonal. Your sentence state otherwise.
Response 5: The sentence has been rewritten by deleting ‘for all three genes’ and ‘with any of the three genes’. A new sentence is ‘Samples with a single parasite genotype were classified as mono-infection while samples with more than one parasite genotype were classified as polyclonal infection’ (see revised manuscript lines 158-159).
Comments 6: Table 2. as mentioned, 146 successfully detected the msp1. This means 33 samples failed. Are they parasite-negative samples? Also, why only consider the alleles frequencies and not showing the allelic families such as MAD20/K1 or MAD20/RO33 combinations. The authors already mentioned in lines 156-157: Samples with a single parasite genotype for all three genes were classified as mono-infections while samples with more than one parasite genotype with any of the three genes were classified as polyclonal infection.
Response 6: All samples included were confirmed to be P. falciparum using qPCR based on Var ATS. For the 33 samples that failed to undergo msp1 amplification, the explanation could be DNA amount not enough to be amplified and/or low parasite density below the detection limit. The sentence with mono and polyclonal infection' meaning had been revised (see revised manuscript lines 158-159).
Reviewer 2 Report
Comments and Suggestions for Authors
This study provides the first characterization of Plasmodium falciparum genetic diversity in the Korhogo district of Northern Côte d’Ivoire, using capillary electrophoresis-based genotyping of the msp1, msp2, and glurp loci from dried blood spot samples collected between 2019 and 2020. The analysis revealed a high number of distinct genotypes and elevated expected heterozygosity (HE ≥ 0.9) across all markers, with two-thirds of infections being polyclonal and higher multiplicity of infection (MOI) and parasite density observed in urban compared to rural areas. These findings highlight intense malaria transmission in Korhogo, provide baseline data for monitoring the impact of control interventions, and underscore the need for further studies employing deep amplicon sequencing to refine understanding of parasite diversity and drug resistance patterns.
While this study offers valuable baseline data on Plasmodium falciparum genetic diversity in Northern Côte d’Ivoire, certain aspects could be improved in future research. The sample set was skewed toward symptomatic patients, limiting the ability to fully assess asymptomatic reservoirs, which are known to contribute substantially to transmission. Inclusion of a larger proportion of asymptomatic carriers would enhance representativeness. Additionally, the analysis did not include detailed characterization of the RO33 allele due to its monomorphic nature, and did not incorporate molecular markers of antimalarial resistance. Employing high-resolution approaches such as amplicon deep sequencing could provide a more granular view of parasite diversity, improve allele detection accuracy, and allow concurrent monitoring of resistance-associated mutations.
Author Response
We would like to thank you for taking the time to review this manuscript. Indeed, you have carefully studied our manuscript according to your constructive comments and suggestions to improve this current paper and futures studies.
Comment and suggestion: This study provides the first characterization of Plasmodium falciparum genetic diversity in the Korhogo district of Northern Côte d’Ivoire, using capillary electrophoresis-based genotyping of the msp1, msp2, and glurp loci from dried blood spot samples collected between 2019 and 2020. The analysis revealed a high number of distinct genotypes and elevated expected heterozygosity (HE ≥ 0.9) across all markers, with two-thirds of infections being polyclonal and higher multiplicity of infection (MOI) and parasite density observed in urban compared to rural areas. These findings highlight intense malaria transmission in Korhogo, provide baseline data for monitoring the impact of control interventions, and underscore the need for further studies employing deep amplicon sequencing to refine understanding of parasite diversity and drug resistance patterns. While this study offers valuable baseline data on Plasmodium falciparum genetic diversity in Northern Côte d’Ivoire, certain aspects could be improved in future research. The sample set was skewed toward symptomatic patients, limiting the ability to fully assess asymptomatic reservoirs, which are known to contribute substantially to transmission. Inclusion of a larger proportion of asymptomatic carriers would enhance representativeness. Additionally, the analysis did not include detailed characterization of the RO33 allele due to its monomorphic nature, and did not incorporate molecular markers of antimalarial resistance. Employing high-resolution approaches such as amplicon deep sequencing could provide a more granular view of parasite diversity, improve allele detection accuracy, and allow concurrent monitoring of resistance-associated mutations.
Response: Thank you for your constructive comments and suggestions to improve future research. It will be considered.
Reviewer 3 Report
Comments and Suggestions for Authors
The manuscript "Genetic diversity of Plasmodium falciparum in Korhogo health district, Northern Côte d’Ivoire" is devoted to the study of the causative agent of malaria in Côte d’Ivoir.Considering the epidemiological significance of malaria, the research topic is certainly relevant. Nevertheless I would like to make two comments on the text of this manuscript. The first (and main) comment concerns the discussion section. It seems to me that it is not enough to simply describe the identified genotypes. Perhaps the manuscript would be significantly improved if the authors described in a little more detail the phylogenetic relationships of the Plasmodium falciparum population circulating in Korhogo with the global population. It would also be possible to dwell in more detail on the question of whether the identified genotypes can be used in epidemiological investigations as markers of the malaria pathogen originating specifically from Côte d’Ivoire.
The second remark concerns the illustrations. Figure 2 is extremely difficult to understand. The need to compare the gradations of one color on the diagram and on its legend causes certain inconvenience to the reader. Perhaps it would be better to choose a different way of presenting the data.
Author Response
We would like to thank you for taking the time to review this manuscript. Indeed, you have carefully studied our manuscript according to your constructive comments and suggestions to improve this current paper and further studies. Please find the detailed responses below and the corresponding revisions/corrections highlighted/in track changes in the re-submitted files.
Comment and suggestion: The manuscript "Genetic diversity of Plasmodium falciparum in Korhogo health district, Northern Côte d’Ivoire" is devoted to the study of the causative agent of malaria in Côte d’Ivoir.Considering the epidemiological significance of malaria, the research topic is certainly relevant. Nevertheless I would like to make two comments on the text of this manuscript. The first (and main) comment concerns the discussion section. It seems to me that it is not enough to simply describe the identified genotypes. Perhaps the manuscript would be significantly improved if the authors described in a little more detail the phylogenetic relationships of the Plasmodium falciparum population circulating in Korhogo with the global population. It would also be possible to dwell in more detail on the question of whether the identified genotypes can be used in epidemiological investigations as markers of the malaria pathogen originating specifically from Côte d’Ivoire. The second remark concerns the illustrations. Figure 2 is extremely difficult to understand. The need to compare the gradations of one color on the diagram and on its legend causes certain inconvenience to the reader. Perhaps it would be better to choose a different way of presenting the data.
Response: I agree with you that phylogenetic relationships and P falciparum originating specifically from Côte d’Ivoire, would contribute to providing more detail on malaria parasites in the area. However, this study is insufficient to deal with the relatedness between P falciparum circulating in Korhogo and other areas, which requires further analysis. These aspects would be considered to improve further studies in the region.
You are also right about Figure 2, which requires a little time for the reader to make the correspondence between the color and legend. This figure has been moved from the main text to the supplementary files. Each pie chart has been shown on a single page, the front has been increased and a comment has been added (see revised manuscript lines 210-211, 214-215, 218, 220, 314-317; Supplementary Figures S1-S5 ).